# Prototype Implementation of a Digitizer for Earthquake Monitoring System

**DOI:** 10.3390/s24165287

**Published:** 2024-08-15

**Authors:** Emad B. Helal, Omar M. Saad, M. Sami Soliman, Gamal M. Dousoky, Ahmed Abdelazim, Lotfy Samy, Haruichi Kanaya, Ali G. Hafez

**Affiliations:** 1Department of Seismology, National Research Institute of Astronomy and Geophysics (NRIAG), Helwan 11421, Egypt; omar.saad@nriag.sci.eg (O.M.S.); mahmoud.salam@nriag.sci.eg (M.S.S.); abdelazim@nriag.sci.eg (A.A.); lotfysamy@gmail.com (L.S.); aligamal@ltlab.com (A.G.H.); 2Electrical Engineering Department, Minia University, Minia 61517, Egypt; dousoky@mu.edu.eg; 3Graduate School of Information Science and Electrical Engineering, Kyushu University, Fukuoka 819-0395, Japan; kanaya@ed.kyushu-u.ac.jp; 4Department of Control and Computer Engineering, College of Engineering, Almaqaal University, Basraa 61014, Iraq

**Keywords:** earthquake monitoring system, three component sensors, digitizer, decimation filter, MiniSEED, GPS, microprocessor, field testing

## Abstract

A digitizer is considered one of the fundamental components of an earthquake monitoring system. In this paper, we design and implement a high accuracy seismic digitizer. The implemented digitizer consists of several blocks, i.e., the analog-to-digital converter (ADC), GPS receiver, and microprocessor. Three finite impulse response (FIR) filters are used to decimate the sampling rate of the input seismic data according to user needs. A graphical user interface (GUI) has been designed for enabling the user to monitor the seismic waveform in real time, and process and adjust the parameters of the acquisition unit. The system casing is designed to resist harsh conditions of the environment. The prototype can represent the three component sensors data in the standard MiniSEED format. The digitizer stream seismic data from the remote station to the main center is based on TCP/IP connection. This protocol ensures data transmission without any losses as long as the data still exist in the ring buffer. The prototype was calibrated by real field testing. The prototype digitizer is integrated with the Egyptian National Seismic Network (ENSN), where a commercial instrument is already installed. Case studies shows that, for the same event, the prototype station improves the solution of the ENSN by giving accurate timing and seismic event parameters. Field test results shows that the event arrival time and the amplitude are approximately the same between the prototype digitizer and the calibrated digitizer. Furthermore, the frequency contents are similar between the two digitizers. Therefore, the prototype digitizer captures the main seismic parameters accurately, irrespective of noise existence.

## 1. Introduction

Seismological networks with interstation distances of a few kilometers are necessary in order to accurately monitor seismicity and comprehend the crustal processes for many disciplines. For example, with the state-of-the-art progresses in oil and gas exploration, recent wide-ranging seismic surveys mainly also depend on data acquisition systems. Moreover, ground vibrations caused by earthquakes or artificial ground vibrations, like those resulting from civil engineering operations, may be transmitted onto the circulating beam by the accelerator parts. The effects of several ground excitation sources on the Large Hadron Collider (LHC) and high luminosity LHC beams have been monitored, analyzed, and examined as discussed in [1]. Lastly, seismic stations around the world have been utilized recently, for example, to assess how humans affect background noise during COVID, examining the impact of the COVID-19 lockdown on ambient noise levels at seismic stations [2].

Seismic waves can be classified into multiple types and have distinct motion patterns. Body waves and surface waves are the two primary categories of seismic waves. While body waves are able to pass through the Earth’s interior layers, surface waves are able to flow like water ripples at the surface. There are two different kinds of body waves: primary waves, also known as pressure waves or P-waves, and secondary waves, also known as shear waves or S-waves. A compression wave is a P-wave. Both liquid and solid materials can support their growth. Shear waves are called S-waves. Only solid materials allow them to spread [3].

The process of seismic recording involves the use of a device that converts seismic energy, which can appear as ground motion or fluid pressure waves, into an electrical impulse. A basic electro-magnetic instrument called a geophone is used to record onshore seismic data or on the sea bottom during marine seismic acquisition [4]. The geophone generates electricity proportional to the Earth’s particle velocity. Current flows from a wire connected to the geophone and passes through an analog-to-digital converter to generate digital data. However, the energy recorded by one geophone is small. Consequently, multiple geophones are arranged and grouped around a central receiver location called a channel. The analog signals generated from a geophone array are added together to form an output signal that is sent to a central receiver.

In particular, the Egyptian National Seismic Network (ENSN) has more than 100 stations distributed around Egypt. The main objectives of the ENSN are recording, monitoring, and analyzing the natural and artificial seismic events. The basic components in each of the ENSN stations are the seismometer and the digitizer. A seismometer is an instrument that responds to ground motions, which may be caused by earthquakes, volcanic eruptions, or explosions. The output of such a device is an analog signal and is converted to digital signal using the digitizer.

ENSN needs to increase the coverage of seismic stations along the active sources to enhance the study of seismic source analysis. However, the implementation of such a system needs a high number of stations, which are costly. Prototype implementation of a digitizer is motivated by these factors: firstly, the performance of the digitizer is enhanced by increasing the signal-to-noise ratio and reducing the computational complexity. Secondly, the overall cost of the digitizer must be reduced. Third, facilities maintenance and system update procedures must be developed.

Digitizer implementation should consider the following requirements. First of all, the analog front end of the data acquisition device should have an analog low-pass filter to prevent aliasing. This step simply attenuates frequency components greater than the Nyquist frequency, prior to sampling by the analog-to-digital converter (ADC). Secondly, the requirements of analog anti-aliasing filters should include sharp roll-off and a flat pass-band, which complicates the design of these filters. Thirdly, the design of analog-to-digital converters (ADCs) is considered a crucial step in order to obtain better anti-aliasing, higher signal resolution, and support different sampling rates. Fourthly, the data format (packets) requires that the instrument have an accurate timing source (i.e., GPS) for time tagging the data prior to compression and transmission. Finally, the process of data formatting should adapt the standard seismological format to be handy for archiving and processing.

The development of seismic-sensing instruments had passed through different stages during the last eighty years. There are five major stages of developments: the electron tube (optical spot recorders), transistors (analog tape recorders), conventional digital seismographs (digital tape recorders), the 16-bit seismographs, and the 24-bit seismographs [5] and the references included in it. There are hundreds of high-resolution digital seismographs that are manufactured on the market by many companies such as Nanometrics, Kinemetrics, RefTek, ION, Sercel, and many others [6,7,8,9,10]. These digital seismographs are targeting the detection of many kinds of seismic signals starting from passive, pico-, nano-, and micro- to strong seismic events. The development of such seismographs is strongly connected with the continuous advances in the seismic data acquisition methods, together with many other technologies. These technologies may include: the progress in electronic technologies [11], seismic exploration updates, intelligent control, computer science, network topologies [12,13], Internet of Things (IoT) [14], signal processing [15,16], and many other disciplines [17,18].

Numerous methods have been developed recently in an effort to improve and enrich seismic instruments. In [18], a centralized seismic data acquisition system is proposed. This system makes joint acquisition of seismic and electrical data acquisition in the same acquisition station. Also, networking is added for remote monitoring using narrow-band internet of things (NB-IoT) technology. Low-cost seismic data acquisition system is proposed in [19]. This prototype captures seismic data using a micro electromechanical system (MEMS)-based accelerometer sensor. The digital data from 24-bit ADC are fed to the microcomputer (Raspberry Pi) for GPS time stamping and converting to standard earthquake data format (MiniSEED). These data are inserted into a ring buffer for data streaming using SeedLink protocol. A multi-channel seismic data acquisition system is proposed in [20,21], adding graphical user interface (GUI) software to manage the system.

On the other hand, wireless seismic data acquisition system is discussed by many researchers. In [22], a wireless a micro-seismic monitoring is proposed using three component geophones for fracturing monitoring of mining activities. Moreover, acquisition from a wireless geophone is achieved in [23] with the addition of reconfigurable antenna for efficient data transmission. Also, in the wireless prototype [24] that is used in building monitoring, the WiFi module is used in wireless communication. The same topology is used in [25] to transmit seismic data generated from the piezoelectric accelerometer.

Recently, the technology progress on sensor and ADCs has facilitated the implementation of low cost and high-resolution seismographs [26,27,28]. In [27,29], the recording system is based on Arduino. Moreover, Raspberry Shake (RS) seismographs are used in [30,31]. Also, a micro electro-mechanical systems (MEMS) analog acceleration sensor is widely used in these systems [32], especially in earthquake early warning systems (EEWS) [33,34]. In [35,36,37], worldwide EEWSs use low-cost MEMS sensors in their systems. Finally, based on specific features of optical fiber, sensors utilizing optical fiber technology (OFT) may offer a different method for earthquake monitoring systems [38].

In this paper, the prototype of a digitizer for an earthquake monitoring system was designed and implemented. This prototype consists of several blocks, i.e., the power source, the analog-to-digital converter (ADC), GPS receiver, and microprocessor. In addition, three finite impulse response (FIR) filters were used to decimate the sampling rate of the input seismic data according to user need. The prototype converts the incoming data into standard seismological MiniSEED format for easy data archiving and streaming. The data are streamed between seismic station and the main center using SeedLink protocol over TCP/IP. A user-friendly interface in which the user can observe the seismic waveform in real time and configure the parameters of the acquisition unit was proposed. Finally, the prototype achieves a reasonable performance compared to calibrated commercial digitizers.

## 2. Materials and Methods

### 2.1. System Wiring Diagram

The system consists of five blocks: sensor, power source, ADC, microprocessor, and GPS. Comprehensively, the seismic acquisition system is divided into five sectors as shown in the wiring diagram in Figure 1, as follows:

First of all, the seismometer (sensor) is an instrument that responds to ground noises and vibrations such as earthquakes and explosions. A three-component sensor has been used to detect the three directions of the Earth’s movement as an input to the system. Three components are used in seismometers to measure simultaneous movement in three directions: up–down, north–south, and east–west. Information on the earthquake is provided by each direction of movement. Furthermore, it can be observed from the three generated seismograms that the S-wave amplitude is greater on the horizontal components and the P-wave appears stronger on the vertical component. Thus, the ground motion cannot be captured by a single seismometer in one direction [3]. The sensor type is a Trillium Compact geophone from Nanometrics company, Kanata, ON, Canada [39], which is a symmetric triaxial sensor. Trillium Compact is a three-component, broadband, and low-noise seismometer. This sensor has a flat response to velocity from 120 s to 100 Hz. The sensor is out of scope, so an already calibrated commercial device is used.

Secondly, the front-end circuit including the analog-to-digital converter (ADC) is used to convert the analog signal from a seismometer to a digital signal for further processing and analysis. The used ADC has a resolution of 24 bits, and is a delta–sigma type. The maximum sampling rate is up to 50,000 samples/second/channel for four channels simultaneously, and the minimum sampling rate is 1.613 kS/s. The ADC input voltage range is ±60 V. Analog-to-digital converters (ADCs) are used by digitizers to transform analog signal samples into digital values. The sampling rate, fs, is the number of samples divided by the interval length, thus, fs = 1/T. The number of bits the ADC utilizes to digitize the input samples is known as its resolution. For an n bit ADC, the number of discrete digital levels that can be represented is $2^{n}$. Resolution determines the precision of a measurement. Thus, measuring small signals is one usage for a high-resolution digitizer. The detailed features of the ADC (NI 9229) from National Instruments, Austin, TX, USA, and more details are explained in [40]. For controlling the NI 9229, a LABVIEW (version 21) driver for controlling the sampling rate and the acquired number of samples was designed.

Thirdly, the GPS receiver provides the proposed system with time and location information. The location and timing information is very important to build the data header in the standard for the exchange of earthquake data (SEED) format. The seismic data from the ADC are stamped with the time and station meta data. The U-Blox ZED-F9P board, u-blox company, Thalwil, Switzerland [41] is used for high data precision from the global navigation satellite system (GNSS). Moreover, the GPS provides the seismic station coordinates (latitude and longitude) to locate the seismic event on the map. The U-Blox ZED-F9P board is multi-band receiver that delivers centimeter-level accuracy in seconds. It provides concurrent reception of GPS, GLONASS, Galileo, and BeiDou due to multi-band RF front-end architecture. ZED-F9P supports a high update rate for highly dynamic applications and is considered as energy-efficient module; more details exist in the datasheet in [42].

Fourthly, the microprocessor is considered the main unit that connects and powers the front-end circuit, and GPS receiver with each other’s using USB ports. Moreover, the GPS information and the extracted data from the ADC are combined via USB ports. Controlling the seismic data is achieved using a novel LABVIEW (version 21) program, which is run in the microprocessor. Moreover, all data that are archived from the digitizer system are stored in the hard disk of the microprocessor to ensure handy data retrieval or data streaming. In this phase, IW32 Single-Board Computer (SBC), fabricated by Winmate company, New Taipei City, Taiwan, is used as microprocessor [43].

Finally, the system has two power sources: 1. solar battery, and 2. power plug. The power adapter socket can feed the system with power directly from electricity (110–240 V) and be converted to 12 V using a power adapter located internally in the system casing. All electronic boards inside the system are energized by 12 V. Also, there are two power adapters to convert 12 V to 5 V and 3.3 V, according to user needs.

### 2.2. Casing Mechanical Design

The casing of the prototype is an essential part of the system. There are many precautions taken into account while making this design, especially considering that this device will be left in the desert for long periods of time, which could approach several years. These precautions are leak-tight design, good thermal conductivity, good protection for the internal components, and good-looking casing for advertising goals. Moreover, casing should include a monitor for the device local setup and monitoring of the seismometer output. Leak-tight design had to be considered to avoid the dust, water, or vapour accumulation on the electronic components inside the casing. This option was studied carefully where there are many ports for input/output and the casing joints as well. In the case of the input/output ports, military-standard connectors are utilized. These connectors have resistivity against any leak. Regarding the casing joints, a standard rubber ring configuration is used in all joints. Thermal conductivity of the casing is an influencing factor for the general safety of the electronic boards inside the casing. The casing is manufactured from aluminum, which has high thermal conductivity to help to dissipate the internal heat to outside of the casing. The interior painting has been carefully selected to be robust against oxidation and have good thermal conductivity.

Figure 2a shows the front panel for the system. The front panel consists of 4 main connectors: sensor, global positioning system (GPS), power, and ethernet. These connectors are military sockets. The sensor connector contains a 26-pin connector that should be connected to seismometer via cable. This connector has availability to take readings from sensors in three directions: north-south (y-axis), east–west (x-axis), and vertical (z-axis). Then, measuring signal is directed to the front-end circuit. GPS connector connects the GPS receiver to the GPS antenna. Power socket is the main supply to the system boards with 12 V DC from solar batteries. Ethernet socket is to enable user to access the prototype locally using web browser or remotely over a TCP/IP connection. Status LEDs function is to give user notifications about power, ethernet, and timing status. Two extra holes are drilled for any future use. From the top view, there is an upper cover for the prototype top part such as projective capacitive (P-CAB) touchscreen display. The enclosure is designed to follow IP67 rating. The prototype top view is illustrated in Figure 2b. The user can configure and monitor ground motion records via P-CAB touch screen. The USB hub is connected directly to the microprocessor for data retrieval. The system has two power sources: 1. solar battery, and 2. power socket. The power adapter socket can feed system with power directly from electricity (110–240 V) and can be converted to 12 V using power adapter that is located inside the system casing. All electronic boards inside the system need 12 V to be turned on.

### 2.3. Data Format

The data are acquired from three channels sensor using the front-end circuit, (NI 9229). The acquired digital data are saved in ASCII file format with an arbitrary header. However, archiving and exchanging seismological time series data should be formatted according to one of the international standards. One of the well-known standards is the standard for the exchange of earthquake data (SEED) format. SEED format contains time series data alongside the related station metadata. These metadata are the network and station information, including the instrument response and the coordinates information. Most of the station metadata are approximately constant and unchangeable for long periods, therefore, the MiniSEED format contains only very limited metadata such as station code and sampling rate. The MiniSEED data format is used for constant data streaming because of the ability to construct large amounts of data by combining small packets together. Additionally, efficient data archiving rather than the complex structure of the full SEED format is used. The ASCII format is converted to MiniSEED format using Obspy, which is a Python framework [44]. The data stream in Obspy is defined using the header. The header contains five main parameters: (1) network name, (2) station name, (3) channel name, (4) the used sampling rate, and (5) the stamped date and time of the first data sample. Finally, the acquired data are encapsulated by the header to represent the data in MiniSEED format.

### 2.4. Data Storage and Streaming

After converting the data to MiniSEED format, they need to be stored and transmitted to the end user. The ring server is designed for streaming data packets based on TCP/IP connection. The ring buffer operation follows a first-in–first-out (FIFO) scheme. Thus, new packets will be transmitted after older packets being transmitted. Although the ring buffer supports many data types, it is preferable to use MiniSEED format to ensure stability and scalability. TCP/IP connections are used for SeedLink communication with default port equal to 18,000. The SeedLink protocol enables robust data communication between remote location (station) and central location where packets have standard packet size of 512-byte MiniSEED records. The SeedLink protocol inserts MiniSEED packets into the ring buffer. The ring buffer is a memory-mapped storage. Finally, programs such as Seiscomp3 or Earthworm in the central hub can connect to a ring server over the SeedLink protocol for data archiving and processing, as shown in Figure 3. Also, data are stored internally in the microprocessor hard drive for data extraction via USB port in case of offline mode.

### 2.5. Decimation Filter

The output sampling frequency is configured according to the user application. This is achieved using a multistage filter design function in LABVIEW (version 21), as shown in Figure 4. The main settings of the filter design are the filter type, number of decimation stages, and decimation factors.

The user can configure the filter specifications including the input sampling frequency, the passband and stopband frequencies, passband ripple, and stopband attenuation. Therefore, the digitizer is scalable to generate a wide range of different rates. Three sequential filters were designed to decimate the input sampling rate to the desired one. According to sampling theory, it is required to sample the signal with a rate that is greater than twice the signal maximum frequency content (Nyquist rate). This prevents overlapping problems (aliasing) between the spectrum of the modulated signal versions. However, achieving the best signal resolution needs the analog signal to be filtered by a low-pass filter before being sampled by the ADC. The requirements of such a filter are very strict such as having a flat pass band and sharp roll-off. The design of this filter with these constraints is very complex and expensive to be implemented. For this reason, the delta–sigma-type ADC uses the oversampling concept. Thus, delta–sigma-type ADC is widely used in the seismology field because of several advantages, such as better antialiasing and higher signal resolution. The main settings of the filter are to decimate the input sampling rate from 25 kHz to 100 Hz using three stages. The factors of each stage are 25, 5, and 2. The passband edge frequency is 40 Hz until it reaches 50 Hz. The passband ripple is 0.01 dB, while the stopband attenuation is 150 dB. The resulting magnitude response for each stage is shown in Figure 5.

### 2.6. Graphical User Interface (GUI)

A user-friendly interface for monitoring and controlling the proposed system has been built using LABVIEW (version 21) program. This graphical user interface (GUI) software is divided into three tabs. Firstly, the waveform tab shows the real-time waveform recorded from three sensor channels, as shown in Figure 6. Secondly, the health tab indicates the GPS information and other information related to the seismic data such as the used sampling rate. The GPS information declares whether the GPS status is locked or unlocked, and the location information includes the latitude and longitude coordinates and the current GPS time and date, as illustrated in Figure 7. Finally, Figure 8 shows the configuration tab, by which the user can control the settings of the parameters of the ADC and the GPS receiver. Thus, the user can configure the system parameters easily. Moreover, the real-time waveform can be monitored for ground motion and the GPS information.

## 3. Results

The prototype digitizer was installed at a seismic station site in the Egyptian National Seismic Network (ENSN), where a commercial instrument had been already installed. This station is located in the central location of Helwan city and streams its data using TCP/IP protocol. Figure 9 shows the prototype installed alongside a calibrated digitizer manufactured by a Canadian company (Nanometrics). This commercial device is called centaur [45], and each digitizer is attached with the same sensor type for comparison. The sensor type is Trillium Compact [39], which is a symmetric triaxial sensor. Trillium Compact is a three-component, broadband, and low-noise seismometer. The sensitivity for the used sensor is 754.3 V.S/m and the dynamic range is equal to 159 dB at 1 Hz. Larger events can be recorded on scale and closer to the source thanks to the exceptionally high clip level of 26 mm/s. The name of the prototype station is test, and the commercial device is named HLW.

Numerous experimental sets of data are acquired and one example is shown in Figure 10. This example demonstrates that the test station records an event in three directions (Z, N, and E). The events occurred on 29 September 2022, at about 08:53:20.

### 3.1. Comparison with Calibrated Digitizer

The centaur and the protype had the same configuration and the same location of installation. Regarding the previous event, the recorded data from the test station (Figure 11a) are compared to recorded data from the centaur station (Figure 11b.) It can be seen that the two waveforms are approximately the same.

Figure 12 displays the event after enlarging the interval of the event only. Certainly, the two waveforms are not totally identical, which is a result of the difference between the used ADC and the multistage decimation filter design. However, the event arrival time and the amplitude are approximately the same. These two parameters are mainly used in the event solution. Furthermore, the two waveforms are drawn in the same figure to determine the deviation between the prototype digitizer and the calibrated digitizer, as shown in Figure 13. This figure illustrates the correlation between the waveform of the two devices in the Z-direction given the same number of samples.

Regarding to frequency domain analyses, the normalized fast Fourier transform (FFT) and spectrogram are investigated in Figure 14 and Figure 15. The frequency distribution of the 29 September 2022, 08:53:20 GMT event is shown in Figure 14a for the prototype station and Figure 14b for the calibrated digitizer. There is a remarkable similarity found. Specifically, the frequency contents have a considerable increase between 0 Hz and 25 Hz, with represented peaks at 1.8 Hz, 2.5 Hz, 7.4 Hz, 9.5 Hz, 12.7 Hz, and 21.9 Hz. Then, frequency values have a constant reduction behavior until 45 Hz.

Figure 15 compares the spectral analysis between the prototype and the commercial digitizers, which is called a spectrogram. The spectrogram is a graph that shows, for a given frequency range, the strength of a signal over time. It displays the energy variation over time and indicates the frequencies where the signal is strongest using a color spectrum. Most of signal power is concentrated from 0 Hz to 25 Hz, with yellow color and a value of about −90 dB, as illustrated in Figure 15a,b. After that the power is decreased in the two figures, with dominant blue color ranging from −130 dB to −155 dB. However, the TEST data for the prototype digitizer after 25 Hz has an increase in signal power by about −20 dB compared to the calibrated digitizer. the probable cause behind this extra power is an unwanted high frequency noise resulting from an electromagnetic interference. The connections between the sensor and the analog-to-digital converter (ADC) should be well shielded. Although there was noise, primary (P) and secondary (S) waves were clearly captured with different amplitudes, and when compared to the data coming from the commercial system, it was discovered to be inconsiderable.

### 3.2. Integration with ENSN Stations

In ENSN, seismologists observe recorded events such as natural and artificial seismic events (blasts) 24/7. No significant local or reginal earthquakes occurred in the same period of testing. Given this situation, the test station is calibrated against the recorded artificial events (explosions). The test station streams the MiniSEED data to the main center in Helwan using SeedLink protocol. The prototype data are integrated together with the ENSN calibrated stations. This integration examines timing and data quality of the prototype, in the case of an occurrence of an event, relative to a group of stations. The calibration process consists mainly of two steps. Firstly, for effective measurements, the test station is neglected from event solution. Secondly, the event solution is repeated, with the test station added to the solution. By comparing the two solutions for the same event, the test station is calibrated. Event solution depends on three major parameters: the location, depth, and origin time. There are two case studies to calibrate the prototype station.

#### 3.2.1. Case Study #1

In this case study, an explosion occurred on 11 September 2022 near Helwan city, as shown in Figure 16. The start time of the event was nearly 7:35:14 in GMT time zone. This event was recorded at Cairo network stations. This network consists of six stations: Kottamia (KOT) station, which is located in the northeast of Cairo. The southwest stations of Cairo are New Saqqara (NSQR), Wadi El-Ryan (RYAN), and Wadi el Natrun (NAT). Finally, the New Ben Suif (NBNS) and Tell El-Amarna (TAMR) stations are located in the west of Cairo.

In order to evaluate the performance of the prototype station (TEST) with the other calibrated stations, TEST station was firstly omitted from the event solution. Initial analysis for the seismic event is performed by determining phase picks. The two main phases are the primary (P) and secondary (S) waves. The resulted event solution without taking TEST station into consideration is shown in Table 1. This solution declares that there is an event with an origin time 07:35:14.48, which is located at latitude (LAT) 29.7978 and longitude (LON) 31.4566. The depth of this blast event is 0.01 km, with a magnitude equal to 2.09 on average. The event solution depends on five P-phases. This solution is used as a reference to evaluate the performance of TEST station. NSQR is considered the nearest station to the event, being a 26 km away.

The event (11 September 2022) was resolved again after adding the TEST station to the solution, as shown in Table 1. TEST station is considered the nearest station, being only 6 km away from the blast event. The event solution location is updated slightly with a LAT = 29.8953 and LON = 31.3855, which is approximately similar to the reference solution. Thus, adding the TEST station enhance the event location. In addition, the event depth is updated to zero, which is more realistic because the explosions do not have depth. Additionally, the TEST station updated the event magnitude to 2.24 on average.

#### 3.2.2. Case Study #2

Another example to validate the performance of the prototype was also an explosion, which occurred on 29 September 2022. The start time of the event was nearly 09:00:32 GMT. This event was recorded in the Cairo network at four stations: Kottamia (KOT), Wadi El Natrun (NAT), Wadi El-Ryan (RYAN), and Suez (SUZ), and the TEST station, as shown in Figure 17.

Similar to the first example, the performance of the TEST station is evaluated with the same strategy. The TEST station is firstly neglected from the event solution. The summary of this solution is calculated as shown in Table 2. This solution declares that there is an event with an origin time of 09:00:32.48 with location coordinates of latitude (LAT) 29.9992 and longitude (LON) 31.4123 using four phases. The magnitude of this blast is 2 on average, with a depth of 1.69 km. The nearest station to that event is KOT station, 26 km away.

The previous event (29 September 2022) was resolved again after adding the TEST station with the four mentioned ENSN stations. Table 2 illustrates the summary of the solution. After adding the TEST station, the location is updated to 29.985 N and 31.399 using five P picks and one S pick. Moreover, the depth is corrected to 0 km with an acceptable root mean square error (RMS) equal to 0.29. Additionally, the magnitude is approximately the same as the magnitude when excluding the TEST station from the solution. Given that the TEST station is 15 km away from the event, it is regarded as the closest station.

## 4. Discussion and Conclusions

Digitizer implementation for seismic data monitoring starts from acquiring data from sensor to streaming the data to the main the main center. To accomplish this task, the analog data should be digitized using the analog-to-digital converter (ADC), stamping the data with GPS time. The data should be archived locally or streamed remotely to a central hub and formatted to standard seismic format. The overall processes should be managed using robust software for smooth system configuration and precise and continuous data acquisition. Finally, the prototype operation should be tested and compared to calibrated digitizers. Prototype calibration was accomplished by comparing the data recording with commercial digitizers in the occurrence of seismic events.

Calibration is performed using either a single station or a network of stations. Firstly, the prototype evaluation is compared with a calibrated digitizer in the same station in time and frequency domains. For the time domain comparison, the prototype is correlated with the calibrated digitizer in the main signal phases such as P and S picks. The normalized fast Fourier transform (FFT) and spectrogram are examined in frequency domain analyses. FFT analysis reveals a significant similarity between the TEST and calibrated digitizer. However, the spectrogram shows excess power in the prototype frequency contents. This extra power demonstrates an existence of unwanted noise attached with main signal components. Connections between the sensor socket and the ADC should be securely insulated to reduce the electromagnetic interference. In future work, noise analysis should be investigated. Secondly, prototype calibration with integration with a network of stations was performed, and two events were discussed. This integration examined timing and data quality of the prototype, in the case of an occurrence of an event, relative to a group of stations. Overall, for two case studies, including the TEST station in the event solution improves the event location, depth, and amplitude. This demonstrates that, despite the existence of noise, P and S waves are clearly captured with distinguishable amplitudes.

In this paper, we design and implement a high-resolution seismic digitizer. The implemented digitizer consists of several blocks, i.e., the power source, the front-end circuit, analog-to-digital converter (ADC), GPS receiver, and microprocessor. In addition, three finite impulse response (FIR) filters were used to decimate the sampling rate of the input seismic data according to user need. The casing of the prototype is designed to resist harsh environmental conditions. The prototype presents the seismic data in standard seismological MiniSEED format for easy data archiving and streaming. The data are streamed between seismic stations and the main center using SeedLink protocol over TCP/IP. This protocol ensures data transmission without loss as long as the data still exist in the ring buffer. The system has a user-friendly interface, which can observe the seismic waveform in real time, and has the ability to configure system parameters. Finally, the prototype achieves a considerable performance when installed in a station within the Egyptian National Seismic Network (ENSN) compared to the calibrated digitizers in different stations. Field test results show that the event start time and the amplitude are approximately the same between the prototype digitizer and the calibrated digitizer. Furthermore, the frequency contents are similar between the two digitizers. Therefore, the prototype digitizer can capture the main seismic parameters accurately irrespective of noise existence.

## Figures and Tables

**Figure 1 sensors-24-05287-f001:**
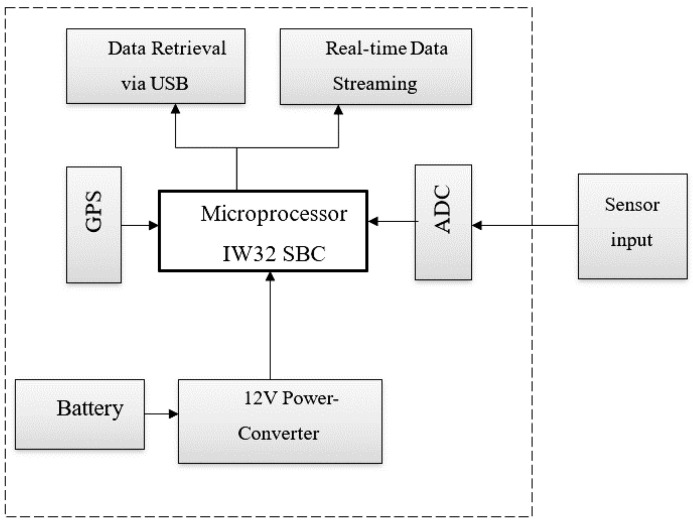
System wiring diagram.

**Figure 2 sensors-24-05287-f002:**
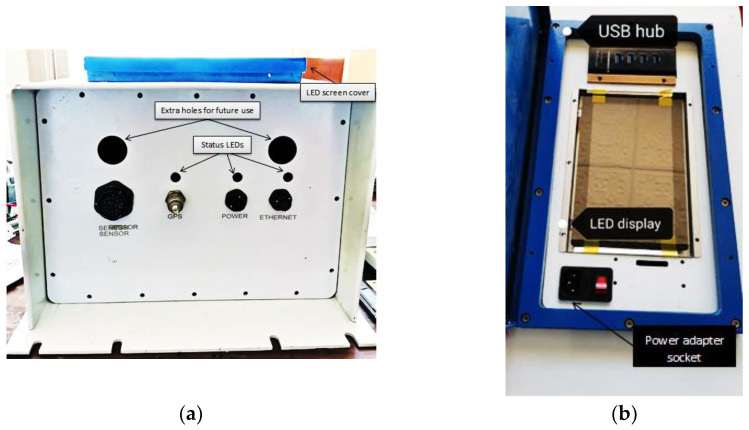
(**a**) System front panel; (**b**) system top view.

**Figure 3 sensors-24-05287-f003:**
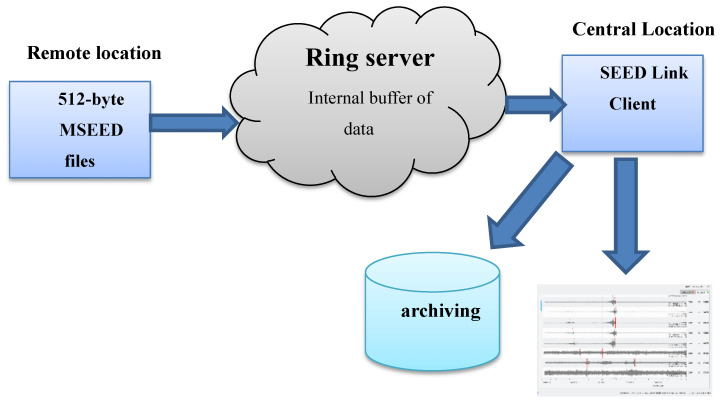
Ring server as a simple SeedLink server.

**Figure 4 sensors-24-05287-f004:**
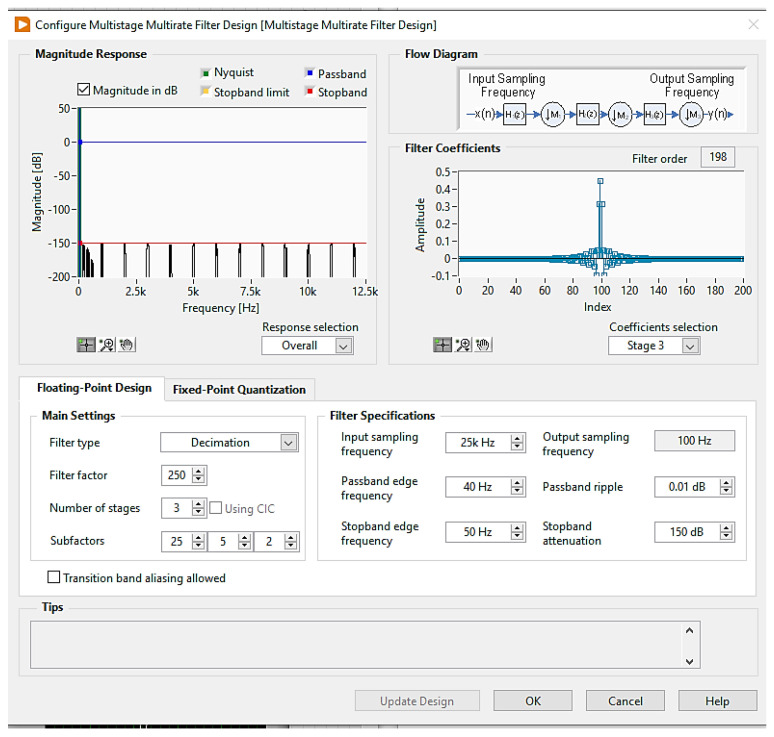
Multistage multi-rate filter design.

**Figure 5 sensors-24-05287-f005:**
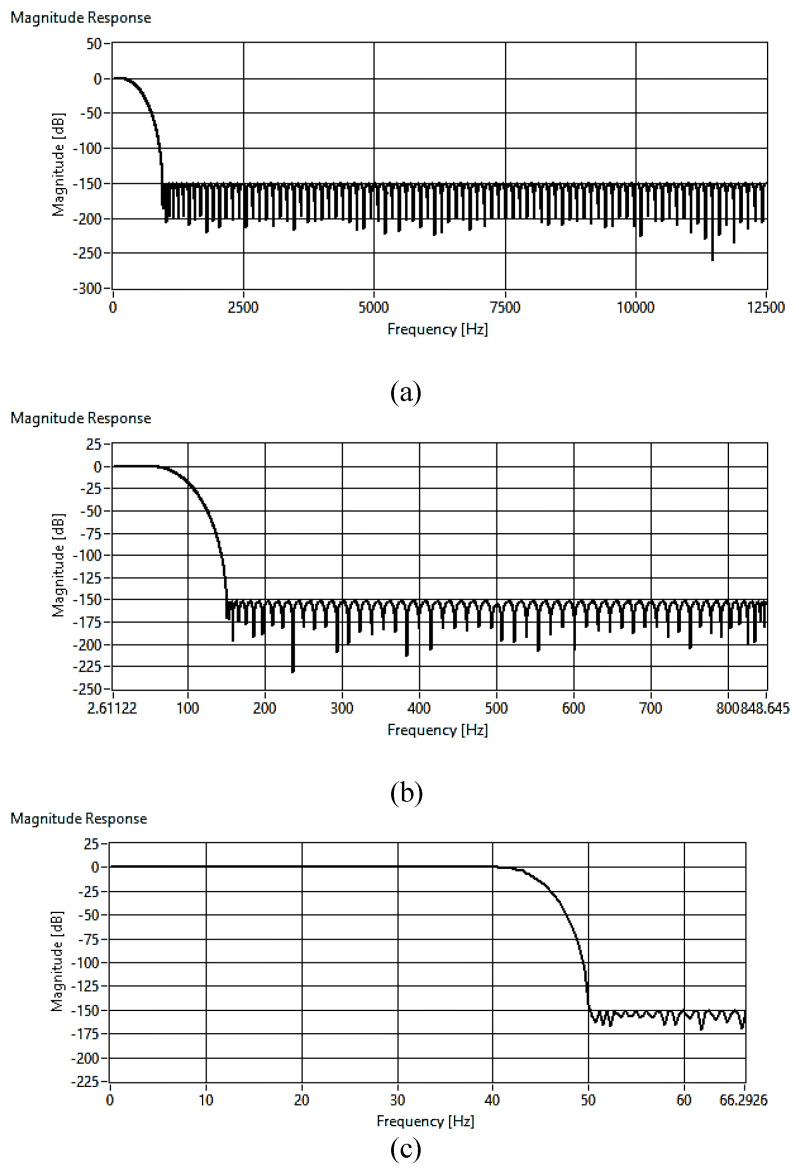
The FIR magnitude response for (**a**) the first stage, (**b**) the second stage, and (**c**) the third stage.

**Figure 6 sensors-24-05287-f006:**
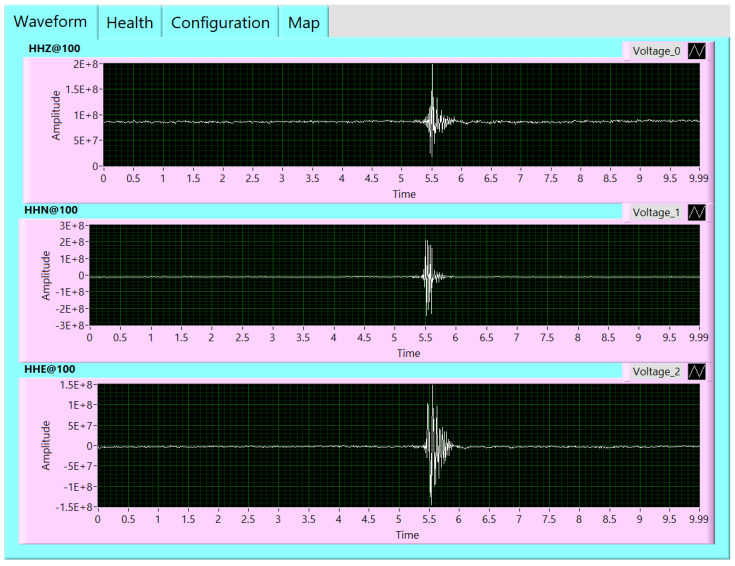
Real-time waveform using the implemented digitizer.

**Figure 7 sensors-24-05287-f007:**
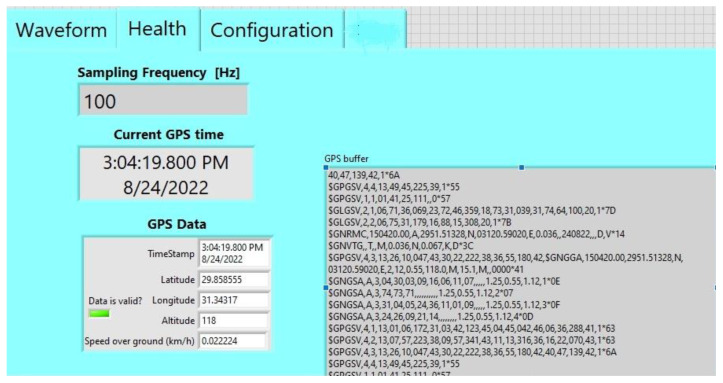
Health tab of the system.

**Figure 8 sensors-24-05287-f008:**
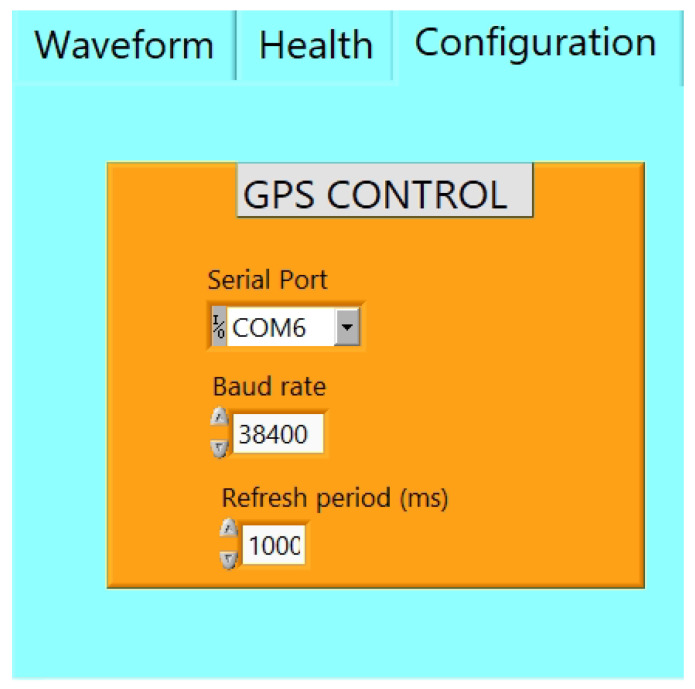
GPS control tab.

**Figure 9 sensors-24-05287-f009:**
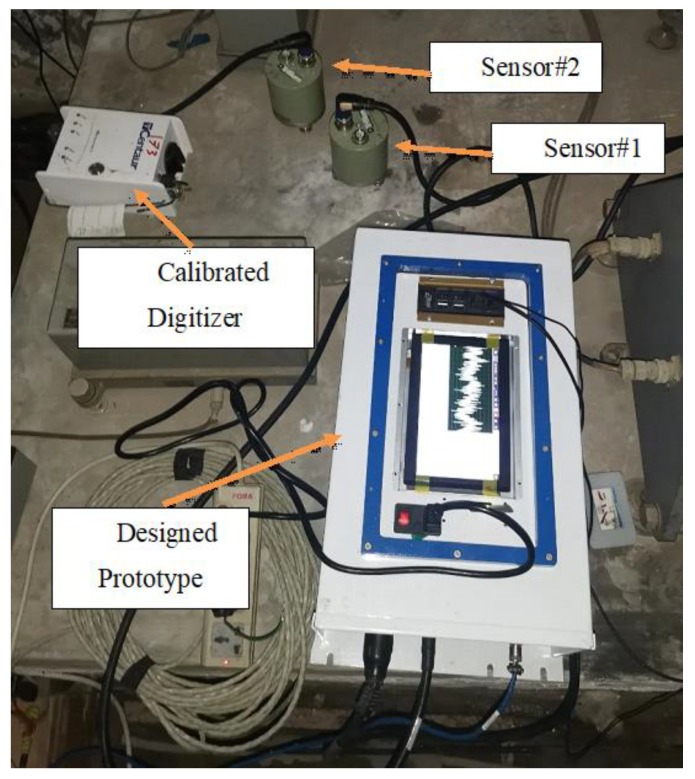
Test station with the calibrated digitizer.

**Figure 10 sensors-24-05287-f010:**
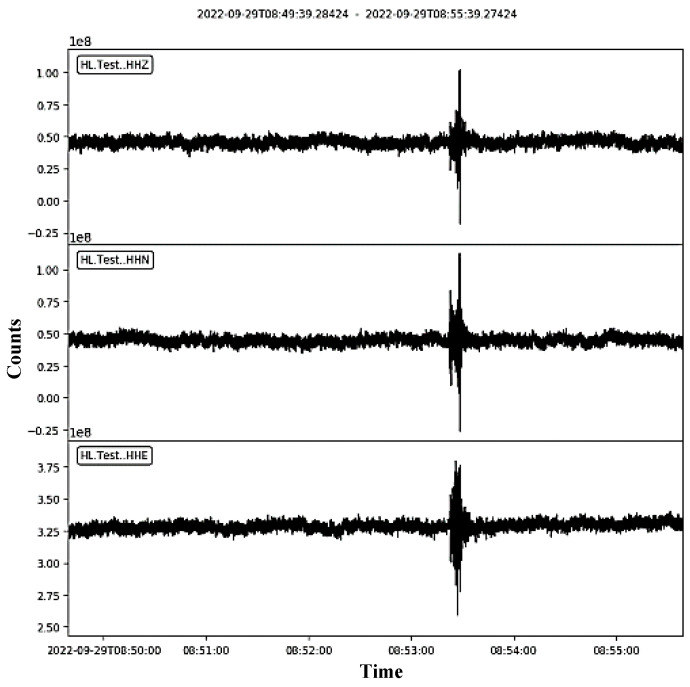
Acquired data from three sensor channels.

**Figure 11 sensors-24-05287-f011:**
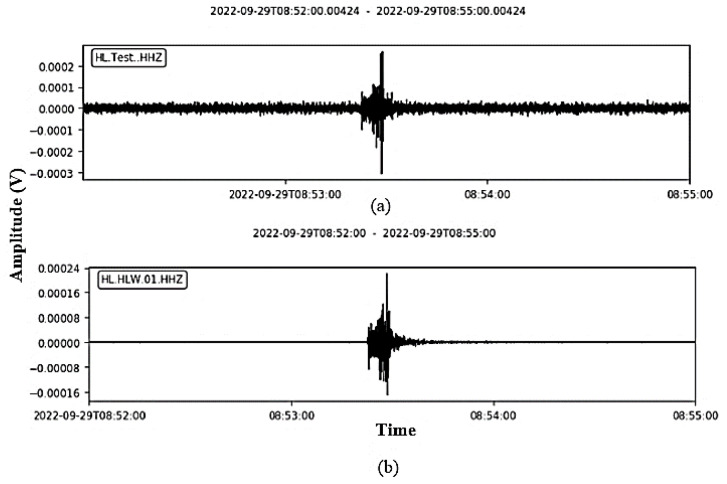
Data comparison between test digitizer (**a**) and centaur digitizer (**b**).

**Figure 12 sensors-24-05287-f012:**
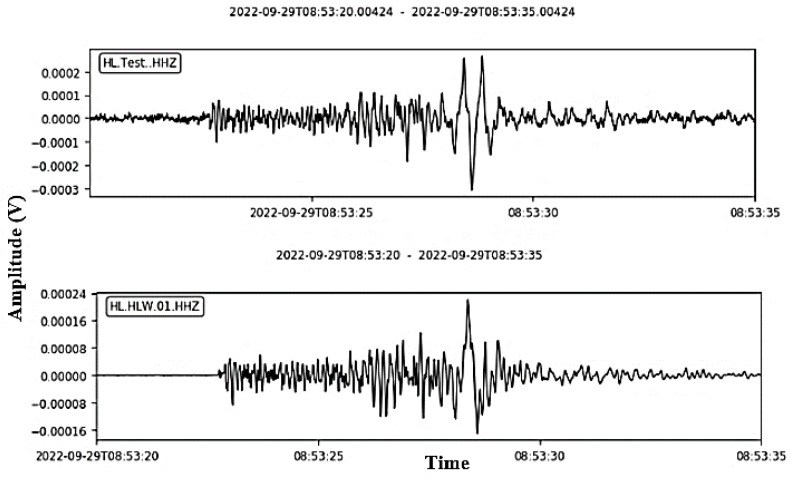
Zooming in on the event interval.

**Figure 13 sensors-24-05287-f013:**
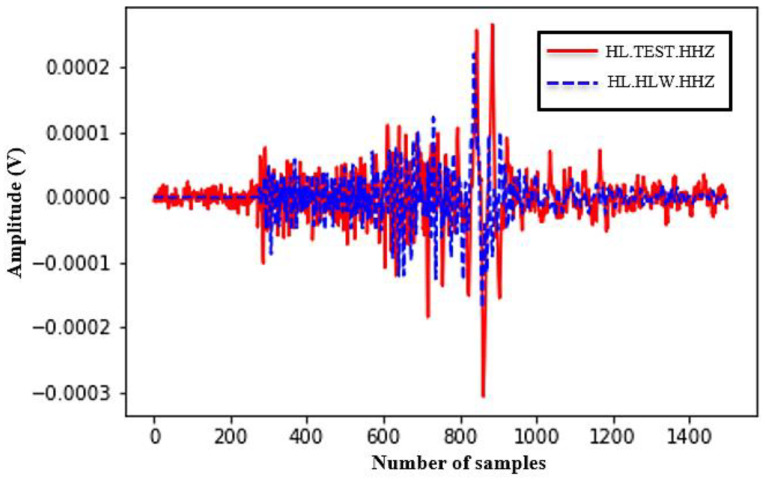
Comparison between the two waveforms.

**Figure 14 sensors-24-05287-f014:**
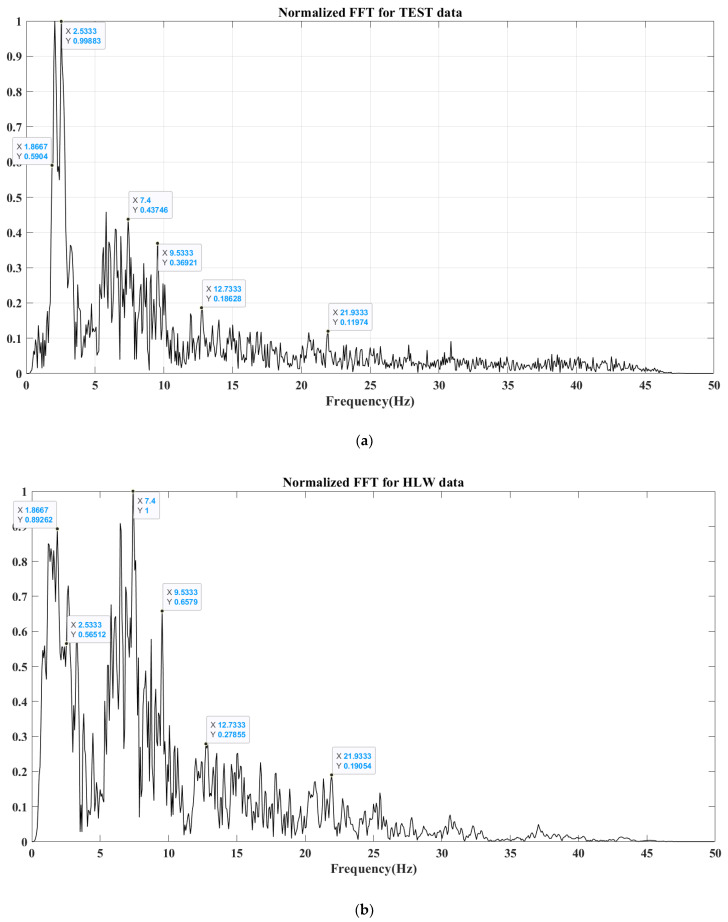
Amplitude spectrum for the prototype (**a**) and calibrated (**b**) digitizers.

**Figure 15 sensors-24-05287-f015:**
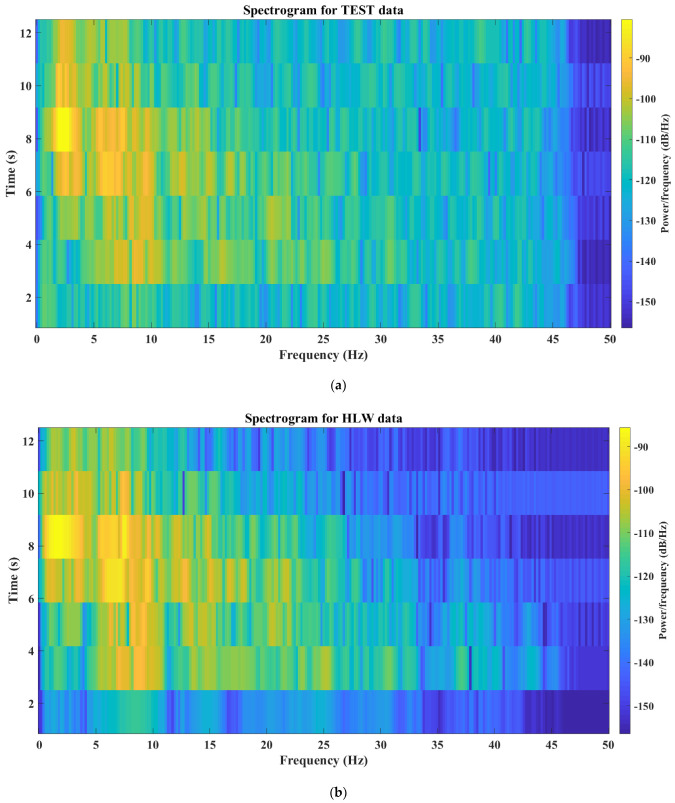
Spectrogram for prototype (**a**) and calibrated (**b**) digitizers.

**Figure 16 sensors-24-05287-f016:**
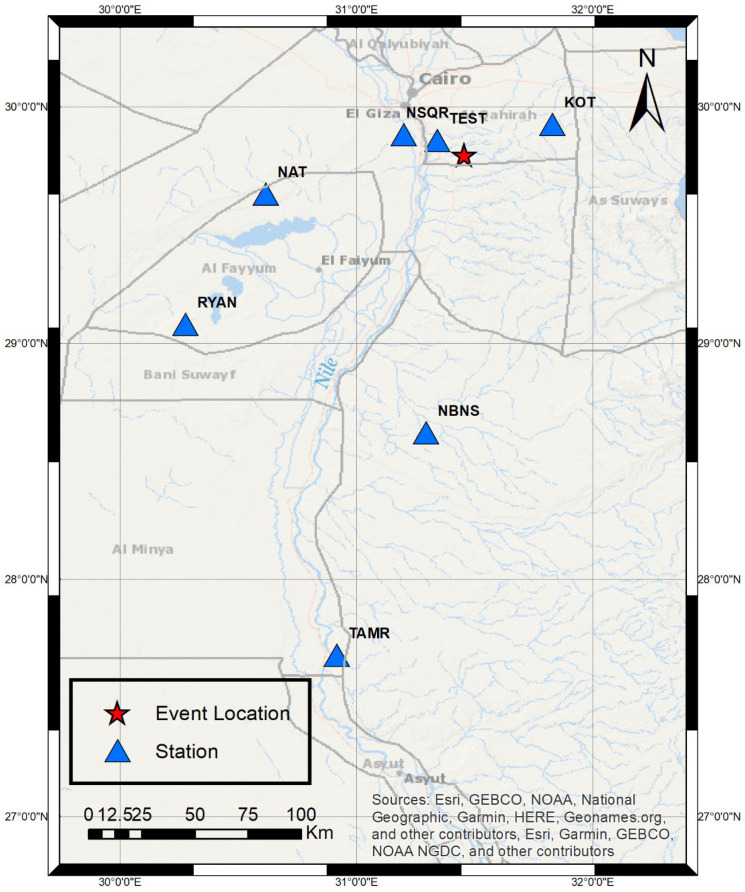
Case study #1 event map.

**Figure 17 sensors-24-05287-f017:**
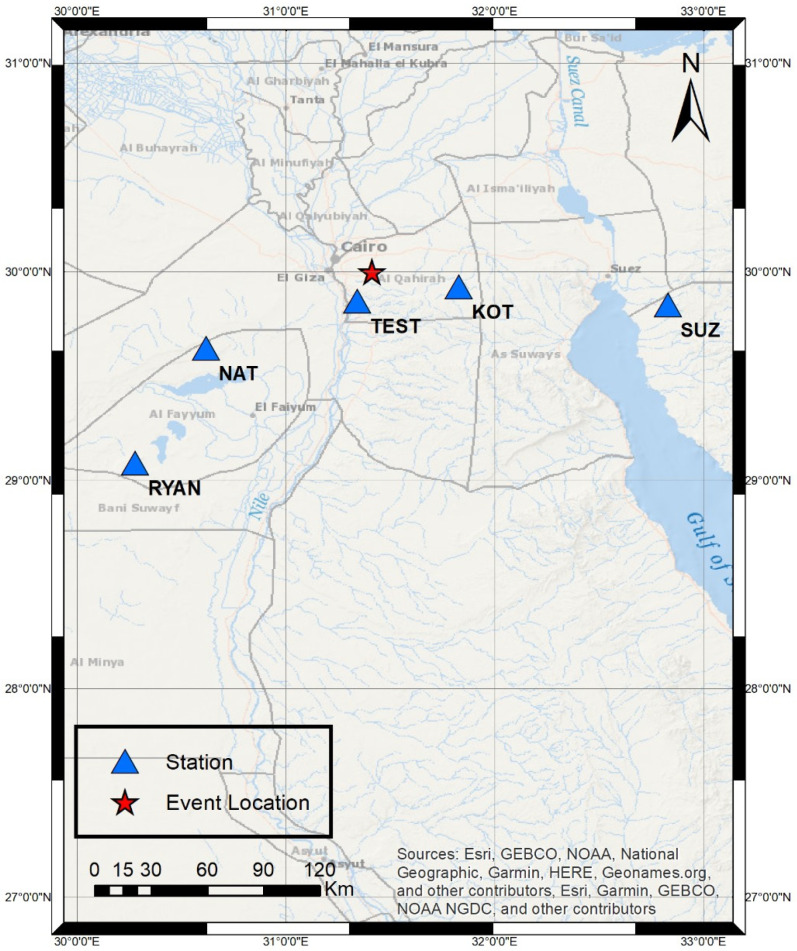
Case study #2 event map.

**Table 1 sensors-24-05287-t001:** Case study #1 solution summary with and without TEST station.

Case Study #1	Date	Time (GMT)	Location		Depth (km)	RMS (s)	Amplitude Magnitude	Phases Used		Nearest Station (km)
			LAT	LON				P	S	
Without TEST St.	11 September 2022	07:35:14.48	29.7978	31.4566	0.01	0.02	2.09	5	0	26
With TEST St.	11 September 2022	07:35:13.86	29.8953	31.3855	0	0.25	2.24	4	0	6

**Table 2 sensors-24-05287-t002:** Case study #2 solution summary with and without TEST station.

Case Study #2	Date	Time (GMT)	Location		Depth (km)	RMS (s)	Amplitude Magnitude	Phases Used		Nearest Station (km)
			LAT	LON				P	S	
Without TEST St.	29 September 2022	09:00:32.48	29.9992	31.4123	1.69	0.00	2	3	1	41
With TEST St.	29 September 2022	09:00:32.05	29.985	31.3997	0	0.29	2.01	5	1	15

## Data Availability

Data are contained within the article.

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
