# Peer review of "Prototype Implementation of a Digitizer for Earthquake Monitoring System"

_sensors, 2024, doi:10.3390/s24165287_

Round 1

Reviewer 1 Report

Comments and Suggestions for Authors

A) General remarks
The research presents in this paper a very interesting topic, as well as results that are of wider significance when it comes to seismic impact assessment and the use of IoT devices. The paper is concise and clear. The literature in the paper is adequately cited. However, some comments on the significance of cited sources will be articulated in the points below.
1.     In the case of literature, all the cited references are relevant to the research. No unnecessary self-citations were detected.
2.    The English language used is without major mistakes, and no significant connections are needed. The reviewer understands that there are different conventions of writing articles in a personal or impersonal form. However, it is more common for scientific papers to use an impersonal form (without “we”, “our” etc”)
3.    The abstract is well written. The role of the abstract is to give a basic overview of the paper. In this case, the abstract provides a good introduction to the form without specific data and is very informative, even for those unfamiliar with the topic reader.  However, the novelty aspects of the paper must be presented in a more robust manner. Similarly, please rethink whether giving information on seismic monitoring system components is necessary in the abstract.
4.    The introduction is mostly well-written and follows all the rules of the proper instruction on the topic. However,
a.    Line 36-52 some statements which are not connected to the literature source.
b.    Lina 37-39 authors mention specific applications for monitoring in the gas and oil industries. If you mention one special case, you should also mention others, e.g. monitoring of ground motion impact on underground research facilities, e.g. https://doi.org/10.1016/j.nima.2023.168495
c. I Would suggest focusing a little more on the background of seismic measurements and in a few lines, introduce typical measurement techniques (seismic station with geophones) before going straight to your solutions. 
d.    point (b) and (c) is also connected to one other issue and special case when the seismic station is used to evaluate the impact of civil engineering works on some sensitive structures like accelerators e.g. investigation and estimation of the LHC magnet vibrations induced by HL-LHC civil engineering activities. Finally, in recent years, seismic stations worldwide were e.g. used to evaluate human impact on background noise e.g. during covid: COVID-19 lockdown impact on CERN seismic station ambient noise levels
5.    Materials and methods. Authors are trying later to validate their systems, but no information is provided on the sensor itself. In the seismic stations, we use geophones and/or strong motion sensors depending on what range we want to analyze, but here, this information, as well as information on sensor parameters, is not provided.
6.    Similarly, in the validation/ results section, I see that the geophones are used for validation on the test site, but no information on those sensors' parameters.
I strongly encourage you to improve the results comparison section, especially the graphical output. I understand what the authors are trying to show,, but how this is presented looks unprofessional.
7.    The paper does provide proper conclusions. However, again, the novelty aspect must be pointed out strongly.
B) Item remarks
Figures are not always of good quality; for some, it is difficult to compare data.
User interface screens are not necessary. They do not give any scientific input to the paper as well as improvements in the graphical improvements of the paper content.
Fig.15 – the waterfall diagram it is really problematic to read due to signal acquisition settings.

C) Conclusions:
The article is clear and easy to follow. Some additional state of the art would be useful, and correction of some editing. The biggest problem is the novelty statement of the paper. Currently, it looks more like an engineering report on the development of seismic digitiser than a scientific paper.

Author Response

Thank you very much for taking the time to review this manuscript. Please find the detailed responses below and the corresponding revisions/corrections highlighted/in track changes in the resubmitted files.

Reviewer 2 Report

Comments and Suggestions for Authors

The suggestion that the system could be solved with a simple IMU raises questions about the necessity and complexity of the digitizer. It would be helpful to explain why a digitizer is required over using an IMU alone. Are there specific features or capabilities that the digitizer offers which are crucial for earthquake monitoring?

The "Materials and Methods" section gives a basic description of the system components, such as the sensor, ADC, microprocessor, and GPS. However, it lacks specific technical details that would help the reader understand the system's functionality better. Providing information like specific model names and technical specifications would be more informative.

The section mentions the use of a three-component sensor but doesn't provide information on why this choice was made or how it enhances the system's performance. It would be helpful to include a brief explanation or reference to relevant literature.

Regarding the ADC's resolution and sampling rate, it would be helpful to explain why these specifications are important for seismic data acquisition and processing. Additionally, citing the source for the ADC model (NI 9229) would give readers a reference point for more information.

The explanation of the GPS receiver is relatively brief. It would be effective to describe why accurate location and timing information are critical for seismic data and provide more details about the U-Blox ZED-F9P board.

The role of the microprocessor in controlling and processing data is mentioned, but the text could provide more information on why the IW32 Single Board Computer was chosen.

The description of the frequency peaks in Figure 14 could be enhanced by explaining the significance of these peaks in the context of seismic monitoring. For instance, what do these peaks indicate about the nature of the event or the sensor's performance?

The mention of primary (P) and secondary (S) waves being "clearly captured with different amplitudes" is positive, but the study could provide more quantitative data to support this claim. How much better or worse were the amplitudes compared to the commercial system?

The study mentions the presence of unwanted noise in the prototype's frequency content but does not delve into the potential sources or implications of this noise. A more detailed analysis of noise sources and potential solutions would be valuable. The conclusion that the noise is "inconsiderable" compared to the commercial system is somewhat subjective. It would be more convincing if supported by specific data or measurements. 

Comments on the Quality of English Language

 Moderate editing of English language required.

Reviewer 3 Report

Comments and Suggestions for Authors

The paper is well described. The language is understandable.

The sections are exhaustive: Methods and Materials, Results, Discussion and Conclusions are clear and show graphs and in-depth design.

I suggest increasing the references in the introduction section by inserting references to seismic fiber optic sensors for example "Opto-mechanical lab-on-fibre seismic sensors detected the Norcia earthquake" https://www.nature.com/articles/s41598 -018-25082-8.

Round 2

Reviewer 1 Report

Comments and Suggestions for Authors

Dear Authors,

All essential and methodological comments have been addressed, and specific improvements have been made.

There are still some minor elements, but mostly of editorial type. Thus, they can be addressed during the next phase.

Currently, I do not have additional requests.

Best regards,

The reviewer.

Reviewer 2 Report

Comments and Suggestions for Authors

Corrections have been made, and it is suitable in its current state, but it requires formatting adjustments.